METHODS

# A multimodal Transformer Network for protein-small molecule interactions enhances predictions of kinase inhibition and enzyme-substrate relationships

Alexander Kroll[1], Sahasra Ranjan[2], Martin J. Lercher[1] *

**1** Institute for Computer Science and Department of Biology, Heinrich Heine University, Düsseldorf, Germany, **2** Department of Computer Science and Engineering, Indian Institute of Technology Bombay, Powai, Mumbai, India

* martin.lercher@hhu.de

**Data Availability Statement:** All datasets are available from Github at https://github.com/AlexanderKroll/ProSmith and from the Zenodo

## Abstract

The activities of most enzymes and drugs depend on interactions between proteins and small molecules. Accurate prediction of these interactions could greatly accelerate pharmaceutical and biotechnological research. Current machine learning models designed for this task have a limited ability to generalize beyond the proteins used for training. This limitation is likely due to a lack of information exchange between the protein and the small molecule during the generation of the required numerical representations. Here, we introduce ProSmith, a machine learning framework that employs a multimodal Transformer Network to simultaneously process protein amino acid sequences and small molecule strings in the same input. This approach facilitates the exchange of all relevant information between the two molecule types during the computation of their numerical representations, allowing the model to account for their structural and functional interactions. Our final model combines gradient boosting predictions based on the resulting multimodal Transformer Network with independent predictions based on separate deep learning representations of the proteins and small molecules. The resulting predictions outperform recently published state-of-the-art models for predicting protein-small molecule interactions across three diverse tasks: predicting kinase inhibitions; inferring potential substrates for enzymes; and predicting Michaelis constants $K_M$. The Python code provided can be used to easily implement and improve machine learning predictions involving arbitrary protein-small molecule interactions.

## Author summary

Understanding how proteins interact with small molecules, such as drugs, is critical to advancing medical, biological, and biotechnological research. Our work introduces Pro-Smith, a machine learning framework that improves the prediction of protein-small molecule interactions. Protein-small molecule interactions can be predicted by using numerical representations of proteins and small molecules as input to machine learning

database at https://doi.org/10.5281/zenodo.
8182031.

**Funding:** This work was funded through grants to MJL by the European Union (ERC AdG "MechSys"–Project ID 101055141) and by the Deutsche Forschungsgemeinschaft (DFG, German Research Foundation: CRC 1310, and, under Germany's Excellence Strategy, EXC 2048/1– Project ID675390686111). The funders had no role in study design, data collection and analysis, decision to publish, or preparation of the manuscript.

**Competing interests:** The authors have declared that no competing interests exist.

prediction models. Previous methods typically generated separate numerical representations for the proteins and small molecules without considering their interactions. ProSmith, however, combines both protein sequence and small molecule structural information in the input of a single multimodal Transformer Network to generate a joint numerical representation. Unlike previous methods, this allows for a comprehensive exchange of information between protein and small molecule, capturing the complex relationships and interactions between these two types of molecules. ProSmith successfully predicts several biological interactions, including kinase inhibitions, potential enzyme-substrate pairs, and enzyme kinetic parameters $K_M$. We provide Python code that can be easily adapted to improve predictions for any protein-small molecule interaction.

This is a *PLOS Computational Biology* Methods paper.

## Introduction

Predicting interactions between proteins and small molecules is a long-standing challenge in biological and medical research, and it plays a crucial role in the discovery of new drugs and in the understanding of their action [1–12]. Moreover, they are essential for predicting enzyme kinetic parameters [13–17] and for inferring diverse protein functions, such as the binding between enzymes and their substrates [18–26].

The main obstacle to achieving such predictions lies in generating effective numerical representations of the two molecule types that encode all the information relevant to the underlying task. Ideally, these numerical representations should already incorporate information on the molecular and functional interactions between proteins and small molecules. Some recent models have made efforts to address this challenge by facilitating the exchange of information between the small molecule representation and the protein representation during their generation [2, 7, 12]. While these approaches offer the potential to better capture the interplay between the two modalities, they still cannot capture the full complexity of protein-small molecule interactions.

Typically, protein information is incorporated into the small molecule representation only after transforming the protein sequence into a single numerical vector, such that no detailed information on individual amino acids is provided. Similarly, small molecule information is presented to the protein representation only after transforming the small molecule information into a single numerical vector that summarizes the properties of individual atoms and their relationships across the whole molecule. This approach results in the loss of information relevant to the downstream prediction task. Instead, one should consider the entirety of both protein and small molecule simultaneously during the creation of their numerical representations. The resulting representation of a protein could be trained to incorporate information specific to small molecules it interacts with according to the training data, while the small molecule representation could integrate information on its protein partners. However, achieving this detailed information exchange poses a challenge, as proteins and small molecules are represented using different modalities. While proteins are commonly represented by their amino acid sequences, small molecules are represented in much greater detail, often as strings containing information about every atom and bond in the molecule [27]. Consequently, separate

deep learning models are typically employed for each molecule type, obstructing effective information exchange.

Popular models for the representation of proteins [28] and small molecules [29] are Transformer Networks, which were originally developed for Natural Language Processing (NLP) tasks. Until recently, Transformer Networks focused primarily on a single input modality, such as text or images. However, the past three years saw the development of multimodal Transformer Networks that project two different modalities onto the same embedding space, facilitating the processing of data that integrates both types of information [30–33]. Notably, such multimodal approaches have achieved significant performance improvements over state-of-the-art methods in tasks involving combinations of images and text.

Inspired by this progress, we here present a novel approach to the prediction of protein-small molecule interactions. Our ProSmith model (PROtein-Small Molecule InTeraction, Holistic model) leverages the power of a multimodal Transformer Network architecture to process protein amino acid sequences and small molecule strings within the same input sequence. For final model predictions, ProSmith combines predictions from three gradient boosting models that utilize (i) a learned representation of the multimodal Transformer Network, (ii) separate general representations of the proteins and the small molecules, and (iii) a combination of all three representations.

This study presents the first multimodal Transformer Network capable of processing molecules with different modalities. The goal of this study is not to provide new biological insights into molecular interactions *per se*, but rather to demonstrate that the developed method can be successfully applied to a variety of prediction problems involving interactions between proteins and small molecules. We show that ProSmith outperforms previous models in predicting protein-small molecule interactions across three diverse tasks: predicting the affinity between protein kinases and drugs [1–12]; evaluating whether a small molecule is a natural substrate for a given enzyme [18–26]; and predicting the Michaelis constant $K_M$ of an enzyme for its substrate [13–17].

## Results

### Architecture of the multimodal Transformer Network for the prediction of protein-small molecule interactions

As its input, the ProSmith model takes two modalities, a protein amino acid sequence together with a SMILES string that represents the molecular structure of the small molecule. When constructing multimodal Transformer Networks, different architectural designs can be considered, each differing in how information between the modalities is incorporated [34]. This subsection provides a detailed description of the architectural choices made in the development of ProSmith.

We adopt a concatenation approach that combines the protein amino acid sequence and the SMILES string into a single input sequence. This choice allows the exchange of all protein and small molecule information at any update step. To process an input, Transformer Networks divide the input sequence into small chunks, referred to as tokens. For protein sequences, each amino acid is treated as a separate token, while the SMILES strings for small molecules are divided into tokens as described in Ref. [29]. To facilitate efficient training, ProSmith leverages pre-learned token representations from Transformer Networks that were trained independently on each modality. For amino acid representations, we utilize embeddings from the ESM-1b model, a 33-layered Transformer Network that was trained in a self-supervised manner on 27 million protein amino acid sequences [28]. For the SMILES string tokens, we use embeddings from the ChemBERTa2 model, a Transformer Network trained on

a dataset of 77 million SMILES strings [29] (see level A in Fig 1). The token embeddings derived from the ESM-1b model and the ChemBERTa2 model have different dimensions, 1280 and 600, respectively. To process these embeddings using the same Transformer Network, we employ linear layers to map both sets of embeddings to a joint embedding space with a shared dimension of 768, which is the hidden dimension of all tokens in the ProSmith Transformer Network (level B in Fig 1).

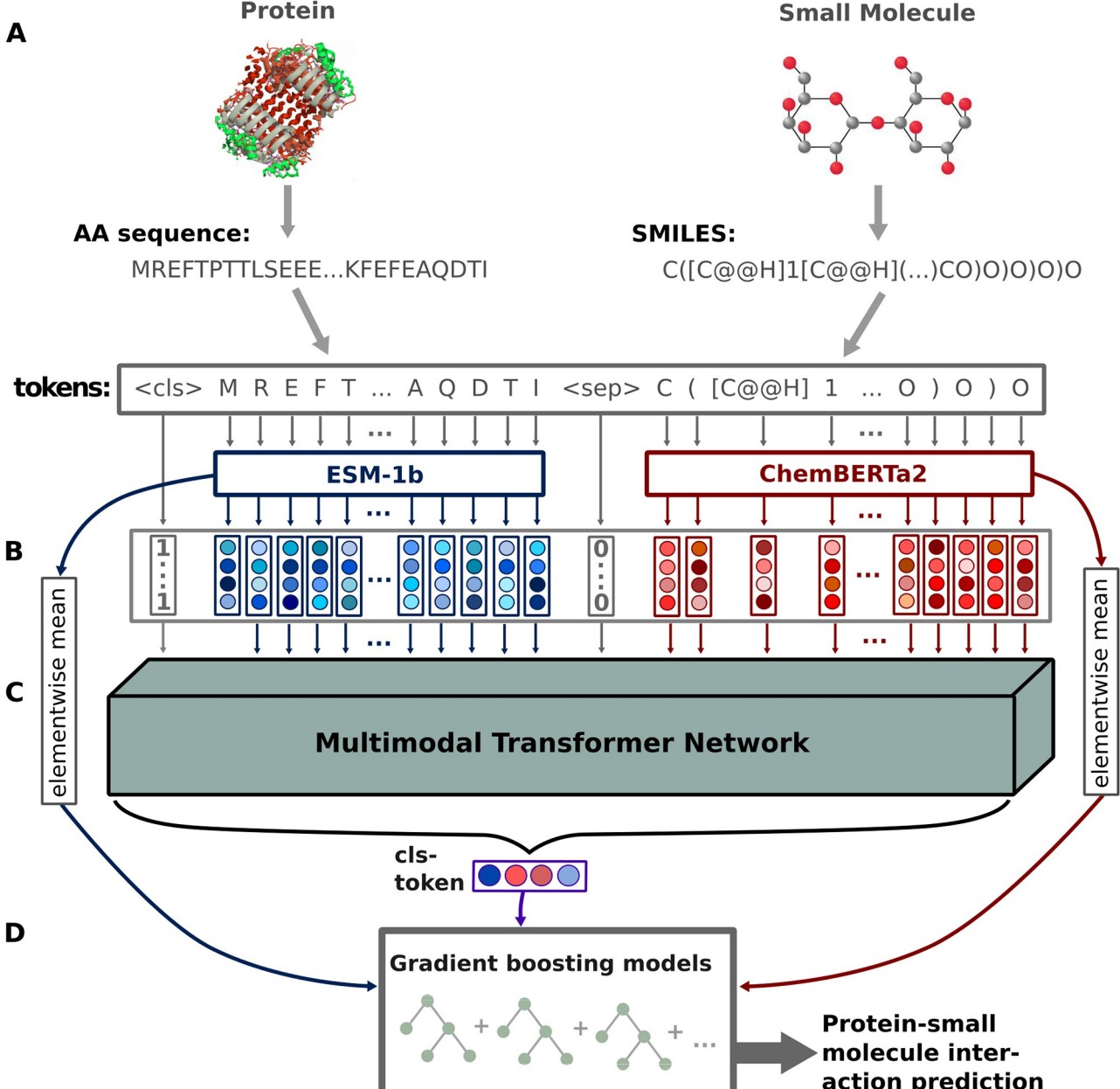

**Fig 1. ProSmith model overview.** In level A, a protein amino acid sequence and a small molecule SMILES string are transformed into input tokens. The protein tokens are converted to embedding vectors using the trained ESM-1b model, while the SMILES tokens are mapped to embedding vectors using the trained ChemBERTa2 model. In level B, all tokens are mapped to the same embedding space, and are utilized as input sequence for a Transformer Network. In level C, the Transformer Network processes the input tokens and puts out an updated embedding of a classification token (cls), which incorporates information from both the protein and small molecule. In level D, this cls vector, in combination with the ESM-1b and ChemBERTa2 vectors, serves as the input for gradient boosting models trained to predict protein-small molecule interactions.

In addition to the protein sequence and the SMILES string tokens, we add two special tokens: the classification token 'cls', its representation is trained as the combined enzyme-small molecule input for downstream tasks, and the separation token 'sep', which is identical across input sequences and indicates to the Transformer Network the end of the protein sequence and the start of the SMILES string within the input sequence (level B in Fig 1). During the processing steps, the Transformer Network (level C in Fig 1) updates each input token using the attention mechanism [35], which enables the model to look at the whole input sequence and to selectively focus only on relevant tokens for making updates. After updating all input tokens for a pre-defined number of steps, the classification token cls is extracted. This token is then used as the input for a fully connected neural network, which is trained to predict an interaction between the small molecule and the protein. By training the entire model end-to-end, ProSmith learns to store all relevant information for the interaction prediction within the cls token.

We set the number of attention layers in the Transformer Network to six, each with six attention heads. In Transformer Networks, the numerical representation of the input sequence is processed in every layer by each attention head separately, updating the input tokens. The updated tokens from all six attention heads are then concatenated and passed as input to subsequent attention layers. The resulting token embeddings from each attention head have a dimension of 128, resulting in the model's hidden dimension of 768.

## ProSmith feeds the learned representations to gradient boosting models

Following the training of the Transformer Network for predicting interactions between proteins and small molecules, we extract the cls token as a task-specific joint representation for a given protein-small molecule pair. However, due to the limited size of the cls token and the number of update steps, we hypothesized that some relevant general information of the protein and the small molecule might be lost during the generation of the representation. To address this concern, we also more directly use the information contained in the ESM-1b representation of the raw protein sequence and the ChemBERTa2 representation of the SMILES string. We create a single representation for a given protein by calculating the element-wise mean [36] across its ESM-1b token embeddings and a single representation for a given small molecule by calculating the element-wise mean across its ChemBERTa2 token embeddings. In the following, we refer to these compressed vectors as ESM-1b vector and ChemBERTa2 vector, respectively.

Previous studies have demonstrated benefits of utilizing learned representations from Transformer Networks as inputs for gradient boosting models, leading to improved outcomes compared to directly using the predictions of a Transformer Network [18, 37]. We thus follow a similar approach here. Gradient boosting models consist of multiple decision trees that are constructed iteratively during training. In the initial iteration, a single decision tree is built to predict a protein-small molecule interaction of interest for all training data points. By constructing new decision trees, subsequent iterations aim to minimize the errors made by the existing trees. Ultimately, an ensemble of diverse decision trees is formed, each focusing on different aspects of the input features and collectively striving to predict the correct outcome [38, 39]. In this study, we leverage the learned cls tokens, ESM-1b vectors, and ChemBERTa2 vectors as inputs for the gradient boosting models.

When aiming to increase model performance, it is a common strategy to train multiple different machine learning models using the same input representations. Using an ensemble of these models, i.e., calculating a (weighted) mean of the various model predictions, can lead to more robust and accurate predictions [40]. We hypothesized that similarly, more robust and

improved predictions can be achieved by an ensemble of the same machine learning model trained with different input representations. In a previous study, we indeed observed that training multiple gradient boosting models with different input vectors and combining their predictions through weighted averaging yielded enhanced performance compared to a single model using all input information simultaneously [15]. Thus, we train three distinct gradient boosting models (level D in Fig 1): one using only the cls token, another using the concatenated ESM-1b vector and the ChemBERTa2 vector, and a third model concatenating all three input vectors. To obtain the final prediction, we compute a weighted mean of the predictions from these models, with the weights determined through hyperparameter optimization.

## Model training and hyperparameter optimization

Each dataset used in this study was divided into three subsets used for training, validation, and testing, respectively. The training and validation sets were utilized for hyperparameter optimizations, where different hyperparameter combinations were used to train the model on the training data, and the set of hyperparameters that yielded the best results on the validation set was selected for the final model. The hyperparameters include learning rate, number of hidden layers, hidden dimension, and batch size (a full list is given in S1 Table).

Due to the substantial time and resource requirements associated with training large Transformer Networks, conducting a systematic hyperparameter search for the multimodal Transformer Network was not feasible on the available hardware. Instead, we employed a trial-and-error approach to identify a suitable set of hyperparameters. We iteratively adjusted the hyperparameters with the aim of improving the results for the drug-target affinity prediction task (see below) on the validation set. The resulting combination of hyperparameters (S1 Table) was used for all tasks in this study. We trained each Transformer Network for 100 epochs. To guard against overfitting, we performed early stopping, i.e., we saved model parameters after each epoch and finally selected the model that achieved the best performance on the validation set.

For the gradient boosting models, we were able to perform a systematic hyperparameter search to identify the optimal configuration for each task. We conducted random searches [41] that iterated through 2 000 combinations of hyperparameters, including learning rate, depth of trees, number of iterations, and regularization coefficients (a full list is provided in S2 Table). After identifying the gradient boosting models that demonstrated the most promising performance on the validation sets, we proceeded to train new models using both the training and validation sets. This final model was then evaluated using the previously untouched test set, ensuring an unbiased assessment of the model's predictive capabilities and ability to generalize.

## ProSmith outperforms previous models for predictions of kinase inhibitions

The process of drug discovery is inherently time-consuming and expensive. A crucial aspect of drug discovery is the determination of interactions between potential drug compounds, typically small molecules [42], and their target proteins. Machine learning models that facilitate large-scale predictions of drug-target affinities (DTAs) have the potential to accelerate the overall drug discovery process by identifying appropriate drug molecules for desired target proteins [43].

Here, we assess the performance of ProSmith on one of the most widely used datasets for validating drug-target affinity prediction models, the Davis dataset [44]. The Davis dataset

comprises 30 056 data points, consisting of binding affinities for pairs of 72 drugs (small molecules) and 442 target protein kinases, measured as dissociation constants $K_d$ (in units of nM). To create target values for ProSmith, we follow previous prediction methods by using log-transformed values, defined as $pK_d = -\log_{10}\left(\frac{K_d}{10^9 \text{nM}}\right)$. The resulting values range from 5.0 to 10.8. Since the Davis dataset consists only of protein kinases, its analysis does not allow direct conclusions about ProSmith's ability to predict drug affinities for other protein families. However, most druggable proteins are kinases [45], and therefore kinase datasets are the most commonly used data for evaluating target inhibition predictions.

To split the Davis dataset, we adopted the identical strategy employed by the previous state-of-the-art method, NHGNN-DTA by He et al. [12]. He et al. split the Davis dataset into 80% training data, 10% validation data, and 10% test data. Four different scenarios were investigated [12]: (i) a completely random split that includes drugs and targets in the test and validation sets that also occurred in other combinations in the training set (random split); (ii) a split that excludes target proteins used for training from the test and validation sets (cold target); (iii) a split that excludes drugs used for training from the validation and test sets (cold drug); and (iv) a split that excludes any drug and any target that were used for training from the validation and test sets (cold drug & target). To obtain accurate estimates of the true model performance, He et al. created five random splits for each of the four aforementioned scenarios. To ensure a fair comparison between ProSmith and NHGNN-DTA, we followed the same procedure, generating the random splits with the code provided in Ref. [12].

The Davis dataset contains only approximately 30 000 data points, which can be considered relatively small. When the available training data for a specific prediction task is limited, a common strategy in deep learning is to pre-train the model on a related task for which more abundant training data is available [46, 47]. To construct such a larger pre-training dataset, we extracted drug-target pairs from BindingDB [48] with experimentally measured IC50 values; these values indicate the concentration of a drug required to inhibit a target by 50%. We excluded all targets and drugs present in the Davis dataset, thereby ensuring that for the cold splitting scenarios, ProSmith has indeed never seen any of the relevant test targets or test drugs before. The resulting dataset comprised approximately one million drug-target pairs with known IC50 values. We pre-trained the ProSmith Transformer Network for six epochs on this dataset (see Methods for additional details). Subsequently, the learned parameters were used as initial parameters to train the ProSmith Transformer Network on the Davis dataset.

We trained ProSmith on the training and validation data of all five random training-validation-test splits for all four splitting scenarios introduced in Ref. [12]. In the following, we state model performance metrics for each scenario as the mean scores resulting from model validation on the five different test sets. To evaluate model performance, we employ performance metrics that have been used widely in previous DTA prediction studies: the mean squared error (MSE); the concordance index (CI); and the $r_m^2$ metric. The CI assesses the ability of a predictive model to correctly rank pairs based on their predicted values. It is defined as the fraction of correctly ordered pairs of predicted values among all comparable pairs in the test set. The $r_m^2$ metric is a commonly used performance metric for quantitative structure-activity relationship (QSAR) prediction models, which penalizes large differences between observed and predicted values. It is defined as $r_m^2 = r^2 \times (1 - \sqrt{r^2 - r_0^2})$, where $r^2$ and $r_0^2$ are the squared correlation coefficients between observed and predicted affinities with and without intercept, respectively [49, 50]. In addition, we also computed coefficients of determination ($R^2$; S3 Table), as $R^2$ is a widely used measure of quantitative prediction accuracy in the machine learning literature.

ProSmith shows improved overall performance compared to previous methods. On the random split (Table 1), ProSmith exhibits a concordance index of CI = 0.911, which is highly similar but slightly lower compared to the previous state-of-the-art method, NHGNN-DTA [12] (CI = 0.914)—thus, of all comparable pairs, NHGNN-DTA ranks 0.3% more correctly. However, ProSmith achieves significant improvements over all previous methods in terms of mean squared error (MSE) and the $r_m^2$ metric. ProSmith is the first method to achieve an MSE below 0.19 on this dataset, lowering the MSE by 0.010 compared to NHGNN-DTA.

In the more practically relevant scenarios where the drug and/or target were not included in the training set, ProSmith also outperforms all previous methods in almost all comparisons (Table 2). In the two scenarios that exclusively contain target proteins not present in the training data (cold target and cold drug & target), ProSmith achieves substantial performance improvements, clearly surpassing all previous models across all three performance metrics. In the cold drug scenario, ProSmith achieves comparable but slightly worse MSE (0.578 vs. 0.554) and CI (0.733 vs. 0.752) values compared to some previous methods. In contrast, ProSmith demonstrates a clear improvement in $r_m^2$ (0.225 compared to the best previous score of 0.207). It is important to note that previous methods did not provide the exact training-validation-test splits, and thus, model performances were not evaluated using the exact same test data. However, since all performance scores result from randomly repeating the same splitting procedure five times, the comparison remains meaningful.

We assessed the predictive capabilities of ProSmith for drugs with different occurrence frequencies in the training set. We generated new training, validation, and test splits, varying the presence of the test drugs in the training set. As expected, model performance improves with increasing occurrence frequency of a test drug in the training data (Fig 2A). Accuracy is low for drugs occurring between 0 and 10 times in the training set. High prediction performance appears to require at least 30 drug-target pairs with the same drug in the training data. This observation contrasts with our finding in a previous study for predicting enzyme-substrate pairs that two training data points with a given substrate already facilitate accurate predictions [18]. Thus, it appears that learning drug-target interactions is much more difficult than learning enzyme-substrate relationships. That enzyme-substrate relationships are easier to learn may be related to the evolution of dedicated binding sites in response to natural selection for the binding of specific substrates, leading to recognizable signatures in the amino acid sequence.

**Table 1. Performance metrics for ProSmith and previously published methods for DTA prediction on the random split of the Davis dataset.** Bold numbers highlight the best performance for each metric. Numbers in brackets indicate the standard deviation across the 5 repeated training runs with different splits. Numbers after the method name show year of publication. Performance scores, except for the results of ProSmith, are taken from Ref. [12]. Arrows next to the metric names indicate if higher (↑) or lower (↓) values correspond to better model performance.

| Method | MSE ↓ | CI ↑ | $r_m^2$ ↑ |
|---|---|---|---|
| DeepDTA (2018) | 0.261 (0.007) | 0.878 (0.002) | 0.63 (0.015) |
| MT-DTI (2019) | 0.245 | 0.887 | 0.665 |
| GraphDTA (2021) | 0.229 (0.005) | 0.893 (0.002) | 0.685 (0.016) |
| GEFA (2021) | 0.228 | 0.893 | - |
| rzMLP (2021) | 0.205 | 0.896 | 0.709 |
| EnsembleDLM (2021) | 0.202 (0.005) | 0.907 (0.004) | - |
| FusionDTA (2022) | 0.208 (0.002) | 0.913 (0.001) | 0.743 (0.002) |
| MgraphDTA (2022) | 0.207 (0.001) | 0.900 (0.004) | 0.710 (0.005) |
| NHGNN-DTA (2023) | 0.196 (0.004) | **0.914 (0.002)** | 0.744 (0.003) |
| ProSmith (this work) | **0.186 (0.003)** | 0.911 (0.004) | **0.760 (0.004)** |

**Table 2. Performance metrics for ProSmith and previously published methods for DTA prediction for different splitting scenarios.** Bold numbers highlight the best performance for each metric under each scenario. Numbers in brackets indicate the standard deviation across the 5 repeated training runs with different splits. Arrows next to the metric names indicate if higher (↑) or lower (↓) values correspond to better model performance. Performance scores, except for the results of ProSmith, are taken from Ref. [12].

| Scenario | Method | MSE ↓ | CI ↑ | $r_m^2$ ↑ |
|---|---|---|---|---|
| Cold target | GraphDTA | 0.510 (0.086) | 0.729 (0.012) | 0.154 (0.014) |
|  | GEFA | 0.433 (0.022) | 0.759 (0.009) | 0.289 (0.016) |
|  | FusionDTA | 0.364 (0.021) | 0.826 (0.011) | 0.435 (0.023) |
|  | MgraphDTA | 0.359 (0.023) | 0.813 (0.008) | 0.425 (0.028) |
|  | NHGNN-DTA | 0.344 (0.029) | 0.855 (0.016) | 0.479 (0.021) |
|  | ProSmith (this work) | **0.294 (0.048)** | **0.870 (0.016)** | **0.602 (0.057)** |
| Cold drug | GraphDTA | 0.920 (0.029) | 0.678 (0.036) | 0.160 (0.019) |
|  | GEFA | 0.847 (0.012) | 0.709 (0.028) | 0.182 (0.015) |
|  | FusionDTA | 0.581 (0.094) | 0.737 (0.012) | 0.187 (0.034) |
|  | MgraphDTA | 0.563 (0.065) | 0.729 (0.022) | 0.192 (0.021) |
|  | NHGNN-DTA | **0.554 (0.091)** | **0.752 (0.017)** | 0.207 (0.030) |
|  | ProSmith (this work) | 0.578 (0.006) | 0.733 (0.027) | **0.225 (0.054)** |
| Cold drug and target | GraphDTA | 0.968 (0.096) | 0.579 (0.017) | 0.026 (0.016) |
|  | GEFA | 0.944 (0.092) | 0.610 (0.029) | 0.032 (0.022) |
|  | FusionDTA | 0.876 (0.091) | 0.645 (0.043) | 0.072 (0.048) |
|  | MgraphDTA | 0.874 (0.090) | 0.636 (0.021) | 0.071 (0.041) |
|  | NHGNN-DTA | 0.857 (0.096) | 0.665 (0.038) | 0.087 (0.051) |
|  | ProSmith (this work) | **0.663 (0.159)** | **0.672 (0.005)** | **0.148 (0.097)** |

We also examined how model performance was influenced by the maximal sequence identity of a test target protein compared to proteins in the training set. This analysis is possible using the cold target data. As expected, increasing protein similarity to training data again leads to improved predictions (Fig 2B). Notably, ProSmith achieves favorable results even for

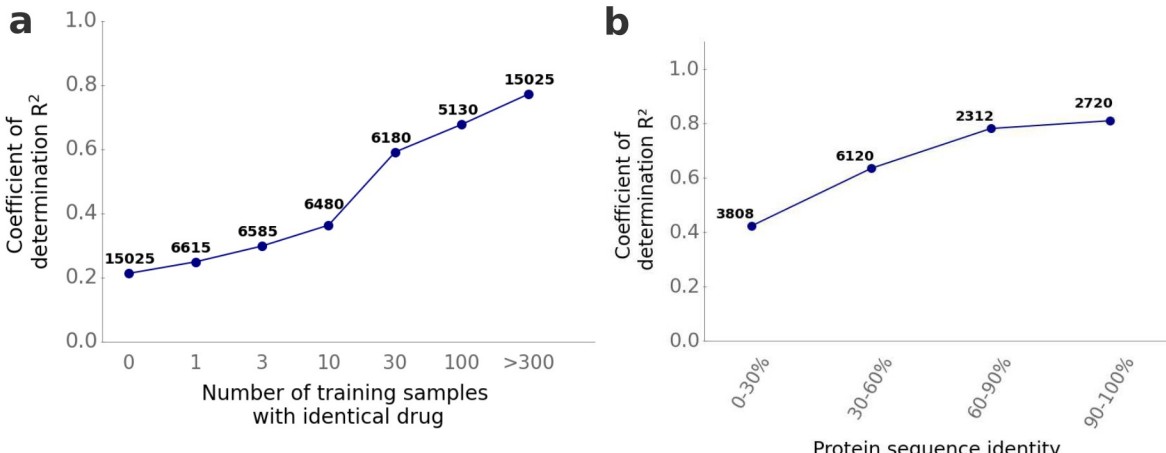

**Fig 2. For accurate DTA predictions, ProSmith requires training on identical drugs but not on similar proteins. (a)** We separately analyzed model performance for dataset splits where drugs from the test set occur in the training for a specified number of times (0, 1, 3, 10, 30, 100, and > 300). We calculated the coefficient of determination $R^2$ for each of those test sets separately. **(b)** We divided all five randomly created test sets under the cold target splitting scenario into subsets with different levels of protein sequence identity compared to proteins in the training set, calculating the coefficient of determination $R^2$ for each subset separately. Numbers above the plotted points indicate the number of test data points in each category.

target proteins that are at most very distantly related to any proteins in the training set (maximal sequence identity < 30%), explaining over 40% of the variance in the target variable. Overall, our findings highlight the superior performance of ProSmith for DTA predictions compared to previous approaches. The results again highlight the model's remarkable ability to generalize to previously unseen proteins.

In addition, we evaluated model performance for test drugs that were not present in the training set based on their maximum similarity compared to all training drugs (S1 Fig). We found that model performance for drugs increases when similar drugs are present in the training set: While model performance is low for drugs with a maximum similarity score below 0.6 compared to any training drug, good prediction results can be achieved for unseen drugs with similarity scores above 0.6. If a test protein was already present in the training set, we also analyzed how model performance changes based on the number of occurrences of the protein in the training set (S1 Fig). However, we did not find a strong correlation between protein occurrence in the training set and model performance. This may not be surprising, since model performance is already high when at least one highly similar protein is present in the training set (Fig 2B).

## ProSmith leads to improved generalization for enzyme-substrate pair prediction

Arguably the most comprehensive high-quality resource of protein sequence and functional information is UniProt [51]. While this database lists over 36 million enzymes, less than 1% of these entries contain high-quality annotations of the catalyzed reactions. Thus, the functions of more than 99% of putative enzymes are currently unknown. To address this challenge, we previously developed ESP, a method that predicts whether a small molecule is a potential substrate for a given enzyme based on the enzyme amino acid sequence and on structural information for the small molecule [18]. The ESP gradient boosting model uses as inputs an enzyme representation from the ESM-1b Transformer model—after task-specific fine-tuning—and a small molecule representation generated through a Graph Neural Network (GNN). ESP, which is currently the only general model for the prediction of enzyme-substrate pairs. By "general model" we refer to a model that, in principle, can be applied to any enzyme without further adaptation. ESP achieves an accuracy of over 91% on this binary classification task. However, the model fails to produce reliable predictions for small molecules that occurred in the training set only once or not at all.

To train and test ProSmith for the same enzyme-substrate prediction task, we use training, validation, and test datasets that are identical to the datasets used in the ESP study. The ESP datasets [18] consist of positive enzyme-substrate pairs with experimental evidence, complemented with sampled negative enzyme-small molecule pairs, with a positive-to-negative ratio of 1 to 3. The dataset was divided into 80% training data and 20% test data, ensuring that no enzyme in the test set has a sequence identity greater than 80% compared to any enzyme in the training set. The training set comprises 55 418 training data points, while the test set contains 13 336 data points. To perform hyperparameter optimization, the training set was further partitioned into 90% training data and 10% validation data.

Given the requirement for a substantial number of data points for training the multimodal Transformer Network, we expanded the training data—but not the test data—by including data with phylogenetic evidence in addition to the data with experimental evidence. This training set, which comprises a total of 850 291 data points, was already utilized in the ESP study to fine-tune the ESM-1b Transformer Network. To reduce the training time associated with this large training set for the multimodal Transformer Network, we increased the batch size from

**Table 3. Performance metrics for ProSmith and ESP for the prediction of enzyme-substrate pairs.** Bold numbers highlight the best performance for each metric. Arrows next to the metric names (↑) indicate that higher values correspond to better model performance.

| Method | Accuracy ↑ | MCC ↑ | ROC-AUC ↑ |
|---|---|---|---|
| ESP | 91.5% | 0.78 | 0.956 |
| ProSmith | **94.2%** | **0.85** | **0.972** |

12 to 24 for this particular task. While we trained the Transformer Network with the expanded training set, we subsequently trained the gradient boosting models using only the smaller training set based on experimental evidence.

The ProSmith results show remarkable improvements over the original ESP model (Table 3). Notably, the accuracy (percentage of correct predictions) increased from 91.5% to 94.2%, the ROC-AUC score increased from 0.956 to 0.972, and the Matthews correlation coefficient (MCC), which measures correlation in binary data [52], increased from 0.78 to 0.85. With these results, ProSmith narrows the gap between the performance of the best available method and perfect predictions by over 30% across all three performance metrics.

A key advancement achieved by ProSmith lies in its ability to produce reliable predictions for small molecules that are not represented multiple times in the training set. As shown in Fig 3A, ProSmith (blue dots) increases the MCC from 0.00 to 0.29 for small molecules not present in the training set and from 0.28 to 0.69 for those present only once (Fig 3A). For test substrates that were not present in the training set, we further investigated how model performance depends on their maximum similarity compared to all training substrates (S2 Fig). As in the drug target prediction task, we find that model performance increases when structurally similar substrates are present in the training set: Model performance is low for unseen substrates with a maximum similarity score below 0.6 compared to all training substrates, but moderate prediction results are obtained for substrates with similarity scores above 0.6.

Furthermore, we investigated the predictive capabilities of ProSmith for enzymes that exhibit different levels of sequence similarity compared to proteins in the training set.

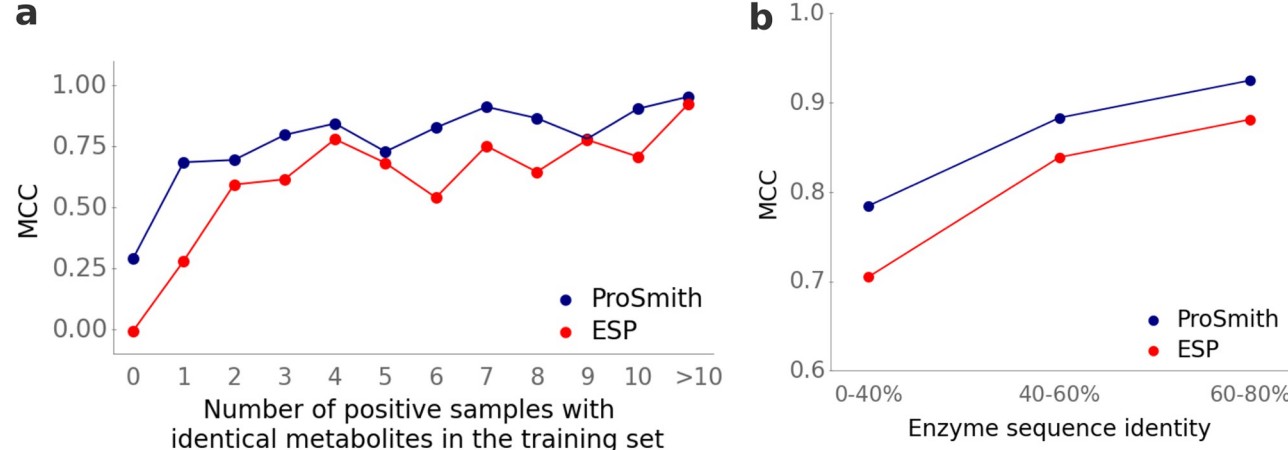

**Fig 3. ProSmith outperforms the ESP model in the prediction of enzyme-substrate pairs especially for molecules with limited representation in the test data.** **(a)** We grouped small molecules from the test set by how often they occur as substrates among all positive data points in the training set, calculating the MCC for each group separately. **(b)** We divided the test set into subsets with different levels of maximal enzyme sequence identity compared to enzymes in the training set, calculating the MCC for each group separately. The numbers of data points within each subset of panel (a) are listed in S4 Table and for panel (b) in S5 Table.

Mirroring the results for small molecules, ProSmith leads to the most significant improvements for enzymes dissimilar to any protein in the training set (Fig 3B): for test enzymes with less than 40% sequence identity to any protein used for training, the MCC improves from 0.70 to 0.78. In sum, ProSmith clearly surpasses the performance of the original ESP model, showing a much better ability to generalize to small molecules and enzymes with limited representation in the training set.

## ProSmith facilitates improved predictions for enzyme-substrate affinities

The third protein-small molecule interaction task that we investigated is predicting the Michaelis constants $K_M$ of enzyme-substrate pairs. $K_M$ represents the substrate concentration at which an enzyme operates at half of its maximal catalytic rate, and thus indicates the affinity of an enzyme for a specific substrate. Knowledge of $K_M$ values is crucial for understanding enzymatic interactions between enzymes and metabolites, as it relates the intracellular concentration of a metabolite to its consumption rate.

For this task, we utilize a dataset containing 11 676 experimental $K_M$ measurements, which we had compiled to develop a previous model that predicts Michaelis constants [14]. We adopted the same split used in that study, which divided the $K_M$ dataset into 80% training data and 20% test data, while ensuring that the same enzyme-substrate pair would not be in the training and test sets. To obtain a validation set, we further split the original training set in the same way into 10% validation data and 90% training data.

Similar to the situation encountered for the DTA prediction task, the number of available $K_M$ data points is relatively small for training the ProSmith Transformer Network. We thus used the enzyme-substrate prediction task for pre-training, i.e., we initialized the ProSmith Transformer Network for $K_M$ with the final parameters from training the model for the enzyme-substrate prediction. This initialization provides a starting point that allows the model to leverage previously learned knowledge. We also tested using the model parameters that resulted from pre-training the ProSmith Transformer Network on the IC50 values, which we used above for the DTA model. However, this led to slightly worse results.

ProSmith demonstrates superior performance compared to two previous $K_M$ prediction models that utilized the same training and test data [13, 14] (Table 4). We cannot compare the MSE between ProSmith and the ENKIE model, as Ref. [13] does not report this metric or the individual predictions for the $K_M$ test set. Similar to what was seen for the other two prediction tasks, ProSmith enhances the ability to generalize to proteins that differ significantly from those in the training set (S3 Fig). However, its capacity to generalize to unseen substrates remains limited and is very similar to the previous state-of-the-art method (S3 Fig).

Although the overall model performance exhibits clear improvement, the magnitude of performance gain is smaller compared to the enzyme-substrate prediction and DTA prediction tasks. We hypothesize that this comparatively small improvement may be related to the relatively low number of training data points in comparison to the other two tasks. We tentatively

**Table 4. Performance metrics of ProSmith and previously published methods for the prediction of Michaelis constants $K_M$.** Metrics were calculated using the same training and test data for all three models. Bold numbers highlight the best performance for each metric. Arrows next to the metric names indicate if higher ($\uparrow$) or lower ($\downarrow$) values correspond to better model performance.

| Method | MSE $\downarrow$ | $R^2 \uparrow$ | Pearson $r \uparrow$ |
|---|---|---|---|
| ENKIE (2022) [13] | - | 0.463 | 0.680 |
| Kroll et al. (2021) [14] | 0.653 | 0.527 | 0.728 |
| ProSmith (this work) | **0.604** | **0.563** | **0.752** |

conclude that ProSmith yields promising results even with small datasets, but its greatest performance gains are observed when applied to larger training datasets.

## Final predictions use complementary information from the multimodal Transformer Network and the protein/small molecule representations

To obtain its final predictions, ProSmith calculates a weighted mean across the results of three distinct gradient boosting models with different inputs. The weights for the weighted mean calculation are treated as hyperparameters, i.e., they are chosen such that they maximize the performance on the validation set for a given task. Fig 4 shows the weights assigned to each model for all investigated prediction tasks; for the DTA prediction, we calculated the mean across all 5 random splits for each splitting scenario.

The model using solely the cls token as its input and the model combining the Chem-BERTa2 and ESM-1b vectors as its inputs have the greatest influence on the final predictions (Fig 4). It is likely that some relevant information from the 1280-dimensional ESM-1b vector and the 600-dimensional ChemBERTa2 vector cannot be captured fully within the 768-dimensional cls token, which also stores information about the protein-small molecule interaction. Adding separate, general protein and small molecule information appears to be advantageous for the generation of more accurate overall predictions. Our findings indicate that the combination of multiple gradient boosting models trained on different input information yields better and more robust performance compared to a single model utilizing all input information (S6 Table), consistent with previous observations [15].

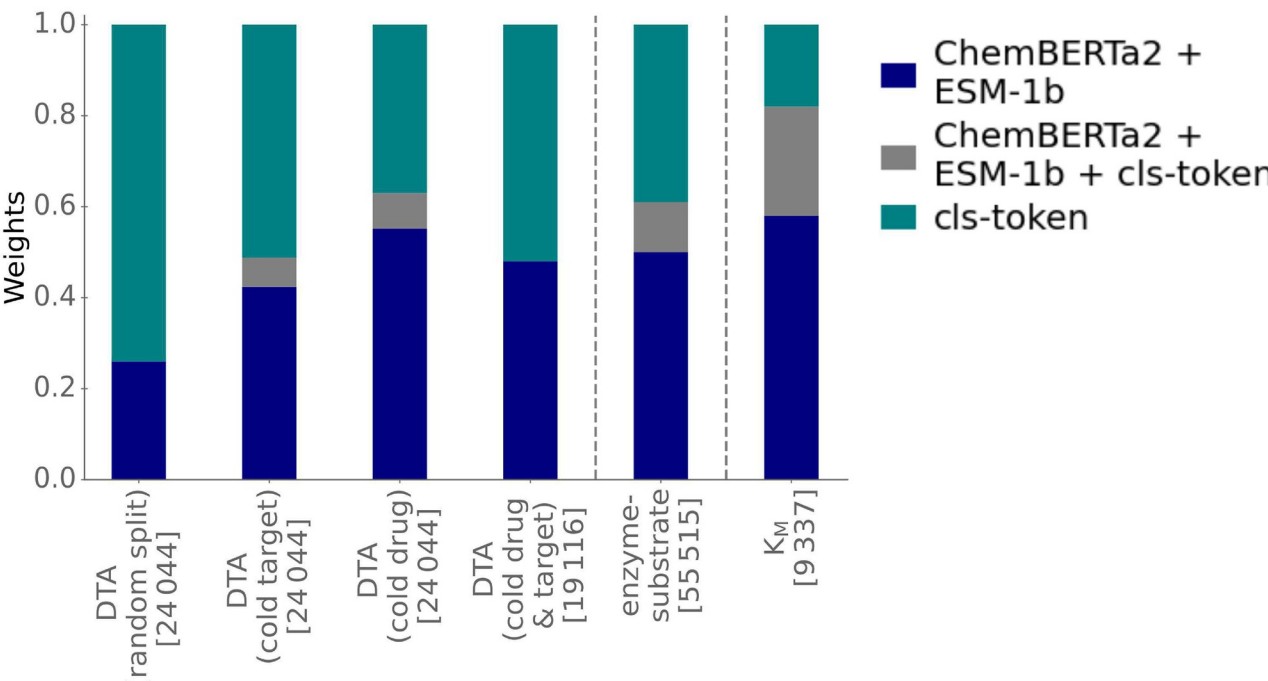

**Fig 4. The optimal ProSmith models combine predictions based on the multimodal Transformer Network with predictions based on separate numerical representations of proteins and small molecules.** The bar plots quantify the weights assigned to the predictions of the three distinct gradient boosting models contributing to ProSmith: the model trained only on the cls token from ProSmith's multimodal Transformer Network (teal); the model combining ESM-1b and ChemBERTa2 vectors (blue); and the model combining all three input vectors (grey). The weights are displayed separately for the distinct prediction tasks: drug-target affinity (DTA) (four different splits); enzyme-substrate pairs; Michaelis constants $K_M$. Numbers in square brackets show the number of training data points.

When predicting Michaelis constants $K_M$, the model utilizing only the cls token exhibits the lowest influence on model performance (Fig 4). This observation might be related to the limited number of only $\sim$ 9000 data points available for training the ProSmith Transformer Network for this task. For tasks with more extensive training data, the influence of the cls token on model predictions is much more substantial. These findings indicate that the ProSmith model can adapt its predictions based on the availability of training data points, optimizing model performance accordingly.

## ProSmith's model architecture has an important impact on model performance

For the ProSmith model, we first train a multimodal Transformer Network end-to-end: input is the protein amino acid sequence and a string representing the small molecule structure, and output is the target feature—e.g., whether protein and small molecule form an enzyme-substrate pair. The final ProSmith model does not use these predictions directly, but instead uses the learned joint protein-small molecule representations to train gradient boosting models. This strategy was motivated by previous studies that showed superior results when adding a gradient boosting step [18, 37]. To investigate whether this additional step indeed contributed to the superior performance of ProSmith, we re-examined the enzyme-substrate prediction task, comparing the model performance of directly using the end-to-end trained multimodal Transformer Network with that of a gradient boosting model that takes the learned joint protein-small molecule representation from this Network as input. The end-to-end trained Transformer network uses a fully connected neural network to process the learned protein-small molecule representations. To facilitate a fair comparison, we explored different numbers of hidden layers and dimension sizes, testing all combinations of one or two hidden layers with hidden dimension sizes of 32, 64, or 128 (S7 Table). In comparison to the full ProSmith model, the best-performing Transformer Network architecture (2 layers with 128 nodes) decreased accuracy from 93.3% to 92.2%, MCC from 0.84 to 0.81, and ROC-AUC from 0.963 to 0.946. Thus, using a gradient boosting model on top of the learned representation indeed improves model performance.

We argue that the main architectural advancement of ProSmith is its ability to process small molecule and protein information in the same input string, which facilitates the exchange of information between protein and small molecule while generating a joint numerical representation. To test whether this is indeed the case and whether the improved performance is not simply due to training a Transformer Network end-to-end, we trained an alternative model for the enzyme-substrate prediction task that uses two separate Transformers to process the protein amino acid sequence and the small molecule SMILES string separately but in parallel. We concatenated the protein and small molecule representations only after they passed through the Transformer Networks, and used the resulting vector as input to a fully connected neural network. As for the ProSmith model, this model was first trained end-to-end before extracting the protein and small molecule representations and using them as input for gradient boosting models. This process is identical to our ProSmith model, except that the model does not allow information exchange between protein and small molecule while generating the numerical representations. Compared to the full ProSmith model, this alternative model decreased accuracy from 94.2% to 93.4%, ROC-AUC from 0.972 to 0.966, and MCC from 0.848 to 0.834. These results indicate that a major architectural advancement of the ProSmith model is indeed its ability to simultaneously process protein and small molecule information in the same input sequence using the multimodal transformer network.

## Discussion

In this study, we introduce ProSmith, a novel machine learning framework for predicting the interactions between proteins and small molecules. The main methodological advance of our model is the utilization of a multimodal Transformer Network that can effectively process amino acid sequences of proteins and SMILES string representations of small molecules within the same input sequence (Fig 1). The ability of this model architecture to incorporate information on the interaction between a protein and a small molecule during the generation of the corresponding numerical representation leads to a superior ability of the trained model to predict drug-target interactions for target protein kinases dissimilar to kinases included in the training data. ProSmith also outperforms previous state-of-the-art methods in predicting protein-small molecule interactions for two different tasks of high relevance to biomedical, biotechnological, and biological research: predicting enzyme substrates and of enzyme-substrate affinities.

Our results highlight the potential of leveraging multimodal inputs to achieve significant advancements in predicting complex molecular interactions. The proposed framework is not limited to modeling protein-small molecule interactions. For example, a very similar approach could be employed to predict protein-reaction interactions, which would be useful for the prediction of enzymatic turnover numbers $k_{cat}$ [15, 16].

A previous study showed that training a specific model for each protein or each small molecule can lead to superior performance compared to a more general model trained for handling different proteins and small molecules [25]. However, such specific models can only be trained for proteins and small molecules for which large amounts of experimental data are available. In contrast, our goal was to develop a model that can generalize to previously unseen proteins, protein families, and small molecules. In the vast majority of test cases in our study, a protein- or small molecule-specific model as described by Goldman et al. [25] cannot be fitted due to a lack of training data. While we have shown in a previous study that a general approach can outperform state-of-the-art models designed specifically for individual enzyme families [18], if one is interested in specific protein families or small molecules with sufficient experimental data, it is conceivable that specific models could lead to superior results.

Due to the computational expenses associated with training Transformer Networks, we did not fully optimize ProSmith's performance for each individual task. Instead, we chose the hyperparameters through trial and error for the drug-target interaction predictions, and we used the same Transformer Network hyperparameters for the two additional tasks. However, hyperparameter search for the Transformer Network, such as optimizing the learning rate, batch size, number of layers, and embedding dimensions, is crucial for improving model performance. In particular, while recent research suggests that some capabilities of Transformer Networks only emerge after surpassing a certain network size limit [53], limited computational resources led us to choose only six transformer layers. An extensive hyperparameter search, executed separately for each individual task, will likely lead to more suitable ProSmith model architectures with improved results.

For the token embeddings of protein amino acid sequences and SMILES strings, we utilized pre-trained embeddings provided by the protein Transformer Network ESM-1b [28] and the SMILES Transformer Network ChemBERTa2 [29]. The parameters of these two networks were not adjusted during the training of ProSmith. In future investigations, it would be valuable to also explore the impact of adjusting the weights of these embeddings simultaneously with the weights of the ProSmith Transformer Network.

Ensemble modeling has been proven effective in enhancing DTA prediction models, as demonstrated in a previous study [9]. Averaging the predictions of multiple well-performing

models, including ProSmith, could yield further performance gains. For instance, while Pro-Smith exhibits a slightly lower concordance index (CI) than the previous state-of-the-art method NHGNN-DTA [12] in the random splitting scenario, the latter shows worse MSE and $r_m^2$ metric scores. Combining these models through a weighted mean prediction approach is likely to overcome the limitations of the individual models, achieving state-of-the-art performance across all three evaluation metrics.

ProSmith appears to show the most substantial performance gains when trained on larger datasets. In deep learning, it is common to pre-train models on similar tasks with more abundant data when training data is limited [46, 47]. We did this for two of the task explored above, pre-training ProSmith on IC50 values before training the DTA prediction model, and on the enzyme-substrate pair data before training the $K_M$ prediction model. Previous studies have shown the benefits of training only the last layers of pre-trained models while keeping the initial layers fixed [47, 54]. Investigating the applicability of this approach to protein-small molecule interaction tasks with small training datasets, such as for the $K_M$ prediction, could be another avenue for future exploration.

The application of the ProSmith framework extends well beyond the three tasks presented in this study. ProSmith can be applied to other protein-small molecule prediction tasks, such as predicting substrates for transport proteins or predicting the activation of proteins through small molecules [55]. Users can employ the Python functions provided on GitHub (https://github.com/AlexanderKroll/ProSmith) to train the ProSmith model for arbitrary protein-small molecule interaction tasks on datasets of up to $\sim$100,000 data points within a reasonable time frame and without the requirement of an extensive GPU infrastructure, as detailed in S1 Text.

## Methods

### Implementation details

All software was coded in Python [56]. We implemented the multimodal Transformer Network in PyTorch [57]. We fitted the gradient boosting models using the library XGBoost [39].

### Calculation of protein token embeddings

We use protein amino acid sequences to represent proteins in the input of the ProSmith Transformer model. Every amino acid in a sequence is represented through a separate token. To numerically encode information about the token, we used learned representations from the ESM-1b model, a Transformer Network with 33 layers that was trained on $\sim$27 million protein sequences [28]. We applied the trained ESM-1b model to each protein amino acid sequence and extracted the updated 1280-dimensional token representations from the last layer of the model.

### Calculation of small molecule token embeddings

We used SMILES strings to represent small molecules in the input of the ProSmith Transformer. To divide the SMILES string into disjoint tokens, we used the ChemBERTa2 model [29]. ChemBERTa2 is a Transformer Network with 3 layers that was trained on $\sim$77 million different SMILES strings. We applied this model to each SMILES string in our dataset, and we extracted 600-dimensional learned token embeddings from the last layer of ChemBERTa2.

## Input representation of the multimodal Transformer Network

Every input sequence of the multimodal Transformer Network has the following structure: first the 'cls' classification token, then the protein amino acid sequence tokens, followed by a separation token, and finally, the SMILES string tokens. In the input, the cls token is represented by a vector of all ones, the separation token is a vector of all zeros, and the protein and SMILES tokens were extracted from ESM-1b and ChemBERTa2, respectively, as described above. The maximum length for protein sequences was set to 1024 and the maximum number of tokens for SMILES strings was set to 256. For longer amino acid sequences, we only kept the first 1024 amino acids; for longer smiles strings, we kept only the fist 256 tokens.

## Model architecture of the multimodal Transformer Network

Before being processed by the multimodal Transformer Network, the amino acid tokens are fed through a protein pooling layer, and the SMILES tokens are fed through a SMILES pooling layer. Each of the two pooling layers is a single-layered fully connected neural network with the ReLU activation function. The pooling layers are applied to each token embedding, mapping the embeddings to the hidden dimension of our multimodal Transformer Network, 768. The parameters of the pooling layers are updated in each iteration of training the Transformer Network.

The classification, protein, separation, and SMILES token embeddings of dimension 768 are used as the input of a Transformer Network called BERT, which stands for Bidirectional Encoder Representations from Transformers [47]. The number of Transformer Layers was set to 6, each with 6 attention heads. The activation function was set to GELU, which is a smoothed version of the ReLU activation function.

After updating each token in the input sequence six times, we extract the updated 768-dimensional representation of the classification token and pass it through a fully connected neural network with one hidden layer of dimension 32 and ReLU as the activation function. The output layer has one node; it uses no activation function for regression tasks and the sigmoid activation function for binary classification tasks.

## Training of the multimodal Transformer Network

We trained the whole model described above end-to-end, i.e., the BERT model together with the pooling layers and the fully connected layers applied to the update classification token. The learning rate was set to $10^{-5}$. The loss function was set to the mean squared error for regression tasks and to the binary cross entropy for binary classification tasks. We trained each Transformer Network for 100 epochs and saved model parameters after each epoch. After training, to guard against overfitting, we selected the model that achieved the best performance on the validation set.

## Processing of batches for the Transformer Network training

Storing all protein sequence tokens and all SMILES string tokens during training requires too much RAM for large datasets. To overcome this issue, we divided the set of all proteins into smaller subsets of size 1000, and we did the same for the set of all SMILES strings. During training, we only load one subset of protein sequences tokens and one subset of SMILES sequence tokens at a time into the RAM, and we iterate over all possible combinations of protein and SMILES subset combinations.

### Pre-training of the Transformer Network on the IC50 dataset

We downloaded the Ligand-Target-Affinity Dataset from BindingDB [48]. We extracted all drug-target pairs with experimentally measured IC50 values from this dataset. We excluded all pairs where either the drug or the target were present in the Davis dataset [44]. This resulted in a dataset with 1 039 565 entries. We split this dataset into 95% training data and 5% validation data. We used the training data to pre-train the ProSmith Transformer Network for the drug-target affinity (DTA) task. We trained the Transformer Network for 100 epochs and saved model parameters after each epoch. After training, we selected the model that achieved the best performance on the validation set. Because of the large training set size, we used a higher batch size of 192 compared to the other tasks investigated in this study. As is common for larger batch sizes, we also increased the learning rate slightly to $1.5 \times 10^{-5}$.

### Splitting the Davis dataset

The Davis dataset consists of 30 056 data points with 72 different drugs and 442 proteins with measured $K_d$ values. To split this dataset into training, validation, and test sets, we adopted the identical strategy employed by the previously leading method, NHGNN-DTA [12]. We generated five random splits for each of four scenarios: random; cold target; cold drug; and cold drug & target (for details, see the section "ProSmith leads to improved generalization for drug-target affinity predictions" in the Results section).

### Splitting Davis data with different occurrence frequencies of test drugs in the training set

To assess the predictive capabilities of ProSmith for drugs with different occurrence frequencies in the training set, we generated new dataset splits. We split the data in such a way that for 15 randomly selected drugs from the test set, only 1, 3, 10, 30, or 100 drug-target pairs with the same drug but paired with different targets are present in the training set. Model performance for drugs that do not occur in the training set or that are present more than 300 times was extracted from the results for the cold drug and random splitting scenario, respectively.

### Training of the gradient boosting models

To find the best hyperparameters for the gradient boosting models, we performed a random grid search with 2 000 iterations. In each iteration, we trained a gradient boosting model with a different set of hyperparameters on the training data and assessed the performance of the resulting model on the validation set. After this random search, we selected the hyperparameter set that led to the best performance on the validation set. We used the Python package hyperopt [58] to perform the hyperparameter optimization for the following hyperparameters: learning rate, maximum tree depth, lambda and alpha coefficients for regularization, maximum delta step, minimum child weight, and number of training epochs. For the task of predicting enzyme-substrate pairs, we added a weight for the negative data points. This hyperparameter was added because the dataset is imbalanced, and it allows the model to assign a lower weight to the overrepresented negative data points during training. We used the Python package xgboost [39] for training the gradient boosting models.

### Computational resources

To train the Transformer Networks and to perform hyperparameter optimization for all gradient boosting models, we used the High Performance Computing Cluster at Heinrich Heine University Düsseldorf (Germany). All training processes were executed on a single NVIDIA

A100 GPU. The only exception was the pre-training of the Transformer Network for the IC50 value prediction. To shorten the training time for the large training set with $\sim$ 1 million data points, we trained this model on four NVIDIA A100 GPUs.

## Calculating small molecule similarities

To compute the maximum similarity of small molecules in the test set to small molecules in the training set, we first compute for each small molecules a binary molecular fingerprint, the extended connectivity fingerprint (ECFP) [59]. ECFPs are 1024-dimensional binary vectors that encode structural properties of small molecules. We then use the Jaccard distance to calculate the pairwise distance between any two ECFPs. The Jaccard distance is defined as the proportion of elements that do not match, considering only those entries where at least one entry is non-zero. The resulting distance measure is a value between 0 and 1, with lower values indicating higher similarity. To convert this distance into a similarity score, we subtracted the distance value from 1 and re-scale all scores so that they range between 0 and 1. This resulted in a similarity score where higher values indicate higher similarity between two molecules.

## Supporting information

**S1 Text. The ProSmith Transformer Network can be trained with limited computational resources.**
(DOCX)

**S1 Table. Hyperparameters of the ProSmith Transformer Network.**
(XLSX)

**S2 Table. Hyperparameters of the ProSmith gradient boosting models.**
(XLSX)

**S3 Table. Coefficient of determination $R^2$ of ProSmith for different splitting scenarios of the Davis dataset.** Numbers in brackets indicate the standard deviation for the results of the 5 repeated training runs with different splits.
(XLSX)

**S4 Table. We divided the ESP test set into subsets according to the number of positive data points with the same small molecule in the training set.** The table shows the number of test data points in each subset.
(XLSX)

**S5 Table. We divided the ESP test set into subsets according to the maximal sequence identity of an enzyme compared to all training enzymes.** The table shows the number of test data points in each subset.
(XLSX)

**S6 Table. Performance metrics for all three trained gradient boosting models with different input vectors and for their combined weighted mean prediction.** For the test sets for the $K_M$ prediction task and for the four splitting scenarios of the Davis dataset, the table lists coefficients of determination $R^2$; for the enzyme-substrate prediction, the table lists Matthews correlation coefficients (MCCs).
(XLSX)

**S7 Table. Replacing the final prediction layer of end-to-end trained Transformer Networks with gradient boosting models improves performance.** The table displays the performance metrics for the end-to-end (E2E) trained Transformer Networks and for a gradient boosting

model for the prediction of enzyme-substrate pairs. The E2E Transformer Network was trained with different numbers of hidden layers and different numbers of nodes in the hidden layers for its fully-connected neural network on top of the attention blocks. The gradient boosting model was trained with the learned joint protein-small molecule embeddings as its only input. Arrows next to the metric names (↑) indicate that higher values correspond to better model performance.
(XLSX)

**S1 Fig. Drug-target affinity predictions improve with increasing drug similarity score compared to the training drugs. (a)** For all cold drug splits, we divided the test set into subsets with different levels of maximum drug similarity scores compared to the drug molecules in the training set. We calculated the coefficient of determination $R^2$ for each group separately. We additionally calculated the coefficient of determination $R^2$ for test drugs that were present in the training set based on the results for the random split scenario. **(b)** For the random split scenarios, we grouped proteins from the test set according to how often they occur as target proteins among all training data points. We calculated the coefficient of determination $R^2$ for each group separately. The numbers above the plotted points indicate the number of test data points in each category.
(EPS)

**S2 Fig. Enzyme-substrate pair predictions improve with increasing substrate similarity compared to training substrates.** For all substrates not present in the training set, we divided the test set into subsets with different levels of maximum substrate similarity scores compared to all substrates in the training set. We calculated the MCC for each group separately. The numbers above the plotted points indicate the number of test data points in each category.
(EPS)

**S3 Fig. ProSmith outperforms previous models in the prediction of Michaelis constants $K_\mathrm{M}$ especially for enzymes not highly similar to proteins in the training set. (a)** We divided the test set into subsets with different levels of maximal enzyme sequence identity compared to enzymes in the training set, calculating the MCC for each group separately. **(b)** We grouped substrates from the test set by how often they occur as substrates among all data points in the training set, calculating the MCC for each group separately. Numbers above the plotted points indicate the number of test data points in each category.
(EPS)

## Acknowledgments

We thank Martin K. M. Engqvist for interesting discussions on multimodal Transformer Networks. Computational support and infrastructure was provided by the "Centre for Information and Media Technology" (ZIM) at Heinrich Heine University Düsseldorf, Germany.

## Author Contributions

**Conceptualization:** Alexander Kroll, Sahasra Ranjan.

**Data curation:** Alexander Kroll, Sahasra Ranjan.

**Formal analysis:** Alexander Kroll, Martin J. Lercher.

**Funding acquisition:** Martin J. Lercher.

**Investigation:** Alexander Kroll, Sahasra Ranjan, Martin J. Lercher.

**Methodology:** Alexander Kroll, Sahasra Ranjan.

**Project administration:** Martin J. Lercher.

**Resources:** Martin J. Lercher.

**Software:** Alexander Kroll, Sahasra Ranjan.

**Supervision:** Martin J. Lercher.

**Validation:** Alexander Kroll, Martin J. Lercher.

**Visualization:** Alexander Kroll, Martin J. Lercher.

**Writing – original draft:** Alexander Kroll.

**Writing – review & editing:** Alexander Kroll, Martin J. Lercher.

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
