## [Decision Letter · Decision Letter 0]

27 Feb 2024

Dear Prof Lercher,

Thank you very much for submitting your manuscript "A multimodal Transformer Network for protein-small molecule interactions enhances drug-target affinity and enzyme-substrate predictions" for consideration at PLOS Computational Biology.

As with all papers reviewed by the journal, your manuscript was reviewed by members of the editorial board and by several independent reviewers. In light of the reviews (below this email), we would like to invite the resubmission of a significantly-revised version that takes into account the reviewers' comments.

Based on the comments of both reviewers as well as my own reading of the manuscript, major revision is required. You should probably see how the method performs on a significant number of other families besides kinases, which might be a particular simple case.

We cannot make any decision about publication until we have seen the revised manuscript and your response to the reviewers' comments. Your revised manuscript is also likely to be sent to reviewers for further evaluation.

Sincerely,

Jeffrey Skolnick

Guest Editor

PLOS Computational Biology

Nir Ben-Tal

Section Editor

PLOS Computational Biology

Based on the comments of both reviewers as well as my own reading of the manuscript, major revision is required. You should probably see how the method performs on a significant number of other families besides kinases, which might be a particular simple case.

Reviewer's Responses to Questions

**Comments to the Authors:**

Reviewer #1: This manuscript proposed a new deep learning model for protein-small molecule interaction prediction. The authors argued that previous methods, which usually combined the representations of proteins and small molecules in the embedding space, might not be able to capture the fine-grained interaction patterns between amino acids and molecular atoms. To address this, they proposed a Transformer-based method that combines the information in the input space, i.e., concatenating the amino acid sequence and SMILES string. The model then relies on the self-attention layer of Transformer to capture the interactions between amino acids and molecular atoms. The proposed method was evaluated on three tasks, including drug-target binding affinity prediction, enzyme-substrate interaction prediction, and enzyme-substrate affinities prediction, with performance improvements over existing methods.

Overall, the manuscript proposed a deep learning model with good results on three different evaluation tasks. I have some major comments about the model design and evaluation, which would help strengthen the work:

Major comments:

- Comparison to per-protein or per-molecule models. A closely related work (Ref 25 in the manuscript) found that training an individual specific model for each protein/molecule had better performance than training a one-for-all model that used a protein-molecule pair as input for all data. Ref 25 also found that a simple method that fits a ridge regression model on ESM embeddings or molecular fingerprints outperformed deep learning-based models. Can the authors compare the proposed model to the per-protein/-molecule models based on ridge regression in Ref 25?

- Investigation of model design. The proposed method used a two-stage approach for the prediction: i) first train a Transformer on the training set, then ii) extract the Transformer’s hidden embedding (the ‘cls’ embedding), combined with ESM and ChemBERTa2 embeddings, to train another gradient boosting model for the final prediction. Could the authors further elaborate/discuss why a Transformer + boosting tree approach was used, instead of an end-on-end approach using the Transformer? The authors cited two papers that showed Transformer + boosting tree had better prediction performance, but I would like to see a more specific discussion on why and in which scenarios (e.g., data size, modality, etc) the Transformer + boosting tree approach would be better than an end-to-end Transformer model? My take from the manuscript is that the Transformer is a good feature extractor, and the gradient boosting tree is a good predictor, but the performance difference between the two-stage and the end-to-end versions of the proposed model is unclear. The authors are suggested to compare with an end-to-end version of their model, e.g., replacing the boosting tree with a multi-layer perceptron (MLP) on top of the embeddings from Transformer, ESM, and ChemBERTa2, and training it end-to-end. It would also be interesting to investigate if varying the number of layers or dimensions of the MLP would lead to a performance close to the gradient boosting tree.

- Ablation study. The authors claimed that concatenating features in the input space is better than concatenating in the embedding space. Although some of the compared baseline methods (e.g., GraphDTA) used the latter approach, those methods did not use the same capable Transformer model. As a more direct test, could the authors compare a variant of their proposed method that combines features in the embedding space, e.g., using two separate Transformers to process the ESM and ChemBERTa embeddings, concatenating them, and then pass them to the predictor?

- Table 3 shows that ProSmith outperformed ESP, but should we attribute the improvement to the ML model or the data augmentation technique? Please clarify with an ablation study, e.g., adding another line in the table of ProSmith’s result without data augmentation.

Minor comments:

- Abstract: “The resulting predictions outperform all previous models” -- I suggest changing “all” to another word since not all existing methods were compared in this work.

- Fig 2: can we also add the leading baselines’ results in this figure? It would better illustrate the performance gap between the proposed method and baselines in the low-data regime.

- Line 244: the authors claimed that their previous ESP model is “the only general model for the prediction of enzyme-substrate pairs”. Could the authors clarify what is a “general model” and why their model is the only general model? How about the models developed in the following PLOS CB paper (cited as Ref 25 in the manuscript): Goldman et al. (2022). Machine learning modeling of family wide enzyme-substrate specificity screens. PLoS computational biology.

- For Figures 2 & 3, can you also show the figures as a function of drug similarity & number of training samples with identical proteins?

- For enzyme-substrate pair/affinity tasks, were the data splits based on random split? How about those cold-start splits used in the drug-target task?

- Table 4: please explain why the MSE value for ENKIE (2022) was missing.

Reviewer #2: To simultaneously process protein amino acid sequences and small molecule strings in the same input, the proposed method map both types of information to the same embedding space, creating input for a transformer network and gradient boosting models. The methods is used for a number of important applications, demonstrating the generality of transformer networks and achieving results that are competitive or better than the ones produced by earlier and more specific methods. These are valuable results.

The paper has clear potential value, assuming the method will be used in the future. Nevertheless, it is disappointing that the entire research is devoid of any biology. To see some interesting applications would substantially increase the value of the contributions. What is particularly disappointing is that the authors refer to the Davis data set (ref 44) as “the dataset comprises 30 056 data 160 points, consisting of binding affinities for pairs of 72 drugs (small molecules) and 442 target proteins, measured as dissociation constants Kd (in units of nM)”. The title of ref 44 is “Comprehensive analysis of kinase inhibitor selectivity”, and the dataset is entirely specific to kinase inhibitors. This has two implications. First, the results are most likely more relevant to these type of interactions rather than generally to protein-drug interactions as suggested by Kroll et al. Second, the results might have provided some interesting insights on the properties of kinase inhibition. However, the word “kinase” is not even mentioned is the Kroll paper. So it seems the authors specifically wants to move away from any implication concerning biology, possibly to imply more generality than provided by the analysis of a kinase-inhibitor dataset. Why not at least mention the specific biology problem, i.e., kinase inhibitor specificity?

While is recognize the contributions of the work for general computer science for demonstrating the application of the same neural network to a variety of problems (which is not entirely novel), I have some doubts whether in its present form this paper makes contributions to computational biology. More generally, it really needs discussing some applications with biological value.

**Have the authors made all data and (if applicable) computational code underlying the findings in their manuscript fully available?**

Reviewer #1: Yes

Reviewer #2: Yes

PLOS authors have the option to publish the peer review history of their article (what does this mean?). If published, this will include your full peer review and any attached files.

Reviewer #1: No

Reviewer #2: No
---

## [Editor Report · Decision Letter 1]

24 Apr 2024

Dear Prof Lercher,

We are pleased to inform you that your manuscript 'A multimodal Transformer Network for protein-small molecule interactions enhances predictions of kinase inhibition and enzyme-substrate relationships' has been provisionally accepted for publication in PLOS Computational Biology.

Best regards,

Jeffrey Skolnick

Academic Editor

PLOS Computational Biology

Nir Ben-Tal

Section Editor

PLOS Computational Biology

Thank you for the revised version which successfully addresses the concerns of the two reviewers. The revised version has been substantially strengthened and improved.

---

## [Editor Report · Acceptance letter]

13 May 2024

PCOMPBIOL-D-24-00044R1 

A multimodal Transformer Network for protein-small molecule interactions enhances predictions of kinase inhibition and enzyme-substrate relationships

Dear Dr Lercher,

I am pleased to inform you that your manuscript has been formally accepted for publication in PLOS Computational Biology. Your manuscript is now with our production department and you will be notified of the publication date in due course.

With kind regards,

Lilla Horvath
